**Trends and spatial shifts in lightning fires and smoke concentrations**
**in response to 21st century climate over the national forests and parks**
**of the western United States**
Yang Li[1], Loretta J. Mickley[1], Pengfei Liu[1], and Jed O. Kaplan[2]
[1]John A. Paulson School of Engineering and Applied Sciences, Harvard University, Cambridge,
MA, USA
[2]Department of Earth Sciences, The University of Hong Kong, Hong Kong, China
*Correspondence to*: Yang Li (yangli@seas.harvard.edu)
**Abstract.** Almost US$ 3bn per year is appropriated for wildfire management on public land in the
United States. Recent studies have suggested that ongoing climate change will lead to warmer and
drier conditions in the western United States with a consequent increase in the number and size of
wildfires, yet large uncertainty exists in these projections. To assess the influence of future changes
in climate and land cover on lightning-caused wildfires in the national forests and parks of the
western United States and the consequences of these fires on air quality, we link a dynamic
vegetation model that includes a process-based representation of fire (LPJ-LMfire) to a global
chemical transport model (GEOS-Chem). Under a scenario of moderate future climate change
(RCP4.5), increasing lightning-caused wildfire enhances the burden of smoke fine particulate
matter (PM), with mass concentration increases of ~53% by the late-21st century during the fire
season in the national forests and parks of the western United States. In a high-emissions scenario
(RCP8.5), smoke PM concentrations double by 2100. RCP8.5 also shows enhanced lightning-
caused fire activity, especially over forests in the northern states.

## 1 Introduction

Both the incidence and duration of large wildfires in the forests of the western United States have increased since the mid-1980s (Westerling et al., 2006; Abatzoglou and Williams, 2016), affecting surface levels of particulate matter (Val Martin et al., 2006; Val Martin et al., 2015), with consequences for human health (Liu et al., 2017) and visibility (Spracklen et al., 2009; Ford et al., 2018). Wildfire activity is influenced by a combination of different factors, including fuel load, fire suppression practices, land use, land cover change, and meteorology (Pechony and Shindell, 2010). Over the forests of the western United States (WUS), lightning-caused wildfires account for the majority of burned area (Abatzoglou et al., 2016; Brey et al., 2018) and have driven most of the recent increase in large wildfires, with human ignition contributing less than 12% to this trend (Westerling, 2016). Studies suggest that a warming climate could enhance wildfires in the WUS (Yue et al., 2013; Abatzoglou and Williams, 2016), but quantifying future wildfire activity is challenging, given uncertainties in land cover trends and in the relationships between fire and weather. Not all these studies that attempt to predict future fire activity have accounted for changing land cover or have distinguished the effects of lightning fire ignitions from human-started fires. In this study, we project lightning-caused fire emissions and wildfire-specific PM concentrations over the national forests and parks of the WUS in the mid- and late- 21$^{st}$ century, using a dynamic global vegetation model combined with a chemical transport model. Our goal is to understand how trends in both land cover and meteorology may affect natural fire activity and smoke air quality over the 21$^{st}$ century.

Consistent with projections of increasing wildfire in the WUS, recent studies have also predicted enhancement of fire-generated PM (smoke PM; BC+OC) under a warmer and drier climate in this region (Yue et al., 2013; Yue et al., 2014; Spracklen et al., 2009; Ford et al., 2018;

Westerling et al., 2006). Some of these studies relied on statistical models that relate
meteorological variables to fire metrics such as area burned; these models can then be applied to
projections from climate models (Yue et al., 2013; Yue et al., 2014; Spracklen et al., 2009;
Archibald et al., 2009; Wotton et al., 2003; Westerling and Bryant, 2008). However, these
statistical methods do not account for changes in vegetation due to climate, increasing atmospheric
$CO_2$ concentrations, or land use. A further weakness of these studies is that they do not consider
whether enhanced fire activity in the future atmosphere may ultimately deplete the supply of
woody fuels (Yue et al., 2013; Yue et al., 2014). Other studies have coupled global vegetation
models to climate models to better represent such fire-vegetation-climate interactions (Chaste et
al., 2018; Ford et al., 2018). Dynamic vegetation models with interactive fire modeling provide
important estimates for long-term and large-scale changes in fire emissions, with most of these
models simulating present-day fire emissions within the range of satellite products but failing to
reproduce the interannual variability (Li et al., 2019; Hamilton et al., 2018). The coupled modeling
approaches integrate vegetation dynamics, land-atmosphere exchanges, and other key physical
processes, allowing consideration of many factors driving fire activity and smoke pollution on
regional scales. Building on this research, we use an integrated vegetation-climate model system
with the aim of clarifying how changing meteorology and vegetation together drive future
lightning-caused wildfire activity. We also provide predictions of smoke pollution at finer spatial
resolution than previously. Our approach accounts for the impact of future climate and lightning
fires on fuel structure, and these fine-scale predictions are of greater utility to environmental
managers and especially the health impacts community.

Lightning is the predominant cause of wildfire ignition in most mountainous and forest

regions of the WUS during months that have high fire frequency (Abatzoglou et al., 2016; Balch
et al., 2017). In remote and mountainous terrain, anthropogenic ignitions are infrequent and >90%
of total area burned is caused by lightning-started fires (Abatzoglou et al., 2016). Here we study
lightning-caused fires over the national forests and parks of the WUS in the mid- and late- 21[st]
century under two future climate change scenarios defined by Representative Concentration
Pathways (RCPs). RCP4.5 represents a moderate pathway with gradual reduction in greenhouse
gas (GHG) emissions after 2050, while RCP8.5 assumes continued increases in GHGs throughout
the 21[st] century. We use the Lund-Potsdam-Jena-Lausanne-Mainz (LPJ-LMfire) Dynamic Global
Vegetation Model (Pfeiffer et al., 2013) to simulate dynamic fire-vegetation interactions under
future climate. LPJ-LMfire, which has been used previously to investigate historical fire activity
(e.g., Chaste et al., 2018), is applied here to estimate natural fire emissions under future climate
simulated by the Goddard Institute for Space Studies (GISS) Model E climate model. July, August,
and September (JAS) are the months of greatest fire activity in WUS forests (Park et al., 2003) and
the focus of our study. We limit the spatial extent of our analyses to the national forests and parks
of the WUS, here defined as 31°N – 49°N, 100°W – 125°W.

**2    Methods**

We quantify the effects of changing climate on area burned and fire emissions caused by

lightning over the national forests and parks in the WUS using the LPJ-LMfire model (Pfeiffer et
al., 2013), driven by meteorological fields from the GISS-E2-R climate model (Nazarenko et al.,
2015). Combined with emission factors from Akagi et al., 2011, dry matter burned calculated by
LPJ-LMfire can be used to estimate natural wildfire emissions of black carbon (BC) and organic
carbon (OC) particles, which are then passed to GEOS-Chem, a 3-D chemical transport model, to
simulate the transport and distribution of wildfire smoke across the WUS. A flowchart of modeling
setup is included in the Supplement (Fig. S1).
**2.1  LPJ-LMfire**

The LPJ-LMfire dynamic vegetation model is driven by gridded climate, soil, land use

fields, and atmospheric $CO_2$ concentrations, and simulates vegetation structure, biogeochemical
cycling, and wildfire (Pfeiffer et al., 2013; Sitch et al., 2003). Wildfires are simulated based on
processes including explicit calculation of lightning ignitions, the representation of multi-day
burning and coalescence of fires, and the calculation of rates of spread in different vegetation types
(Pfeiffer et al., 2013). LPJ-LMfire calculates fire starts as a function of lightning ground strikes
and ignition efficiency. Not every lightning strike causes fire. The model accounts for the
flammability of different plant types, fuel moisture, the spatial autocorrelation of lightning strikes,
and previously burned area. As fires grow in size, the likelihood of fire coalescence or merging
increases. Fires are extinguished by consuming the available fuel or by experiencing sustained
precipitation (Pfeiffer et al., 2013). Our study does not consider changes in human-caused fires,
including agricultural fires.

The climate anomaly fields from the GISS-E2-R climate model used to prepare a future

scenario for LPJ-LMfire are monthly mean surface temperature, diurnal temperature range (i.e.,
the difference between monthly mean daily maximum and daily minimum temperatures), total
monthly precipitation, number of days in the month with precipitation greater than 0.1 mm,
monthly mean total cloud cover fraction, and monthly mean surface wind speed. This version of
the GISS model was configured for Phase 5 of the Coupled Model Intercomparison Project
(CMIP5) (Nazarenko et al., 2015). For RCP4.5, the GISS model predicts a statistically significant
increase in surface temperature of 1.4 K averaged over the entire region by 2050 during JAS; for
RCP8.5, the mean JAS temperature increase is 3.7 K by 2100. In both future climate scenarios,

115 significant precipitation decreases of ~20% by 2100 are simulated. Several studies have predicted

116 future increases in lightning due to climate change (e.g., Price and Rind, 1994a, Romps et al.,

117 2014). However, the relationship between lightning flash rate and meteorology is poorly

118 constrained in models and depends largely on physical parameters such as cold cloud thickness,

119 cloud top height, or convective available potential energy. In our study, lightning strike density for

120 application in LPJ-LMfire is calculated using the GISS convective mass flux following the

121 empirical parameterization of Magi, 2015. Although observations suggest a link between aerosol

122 load and lightning frequency (e.g., Altaratz et al., 2017), we do not consider that relationship here.

123 Unlike surface temperature and precipitation, we find that average lightning density over the West

124 does not change significantly during the 21[st] century, as described in Fig. S2. LPJ-LMfire scales

125 lightning flashes to cloud-to-ground lightning strikes, which are the portion of total flashes in

126 clouds that directly causes natural wildfires (Pfeiffer et al., 2013). Therefore, cloud-to-ground

127 lightning frequencies are also considered constant during the 21[st] century. We run LPJ-LMfire on

128 a 0.5°×0.5° global grid, though for this study only results over the national forests and parks of

129 the WUS are analyzed.

130  The GISS-E2-R meteorology used here covers the period 1701-2100 at a resolution of 2°

131 latitude x 2.5° longitude. The start year of the two climate scenarios, RCP4.5 and RCP8.5, is 2006.

132 The two RCPs capture a range of possible climate trajectories over the 21[st] century, with radiative

133 forcings at 2100 relative to pre-industrial values of +4.5 W m$^{-2}$ for RCP4.5 and +8.5 W m$^{-2}$ for

134 RCP8.5. From 2011 to 2015, the greenhouse gas concentrations of the two scenarios are nearly

135 identical. To downscale the GISS meteorological fields to finer resolution for LPJ-LMfire, we first

136 calculate the 2010-2100 monthly anomalies relative to the average over the 1961-1990 period, and

137 then add the resulting timeseries to a high-resolution observationally based climatology at 0.5°

latitude × 0.5° longitude spatial resolution. The climatology was prepared using the datasets
including WorldClim 2.1, Climate WNA, CRU CL 2.0, Wisconsin HIRS Cloud Climatology, and
LIS/OTD, as described in Pfeiffer et al., 2013. For each RCP, LPJ-LMfire simulates vegetation
dynamics and fire continuously for the period 1701-2100, with monthly resolution. Continuous
400-year simulations allow for sufficient spin-up. The LPJ-LMfire simulations used here cover the
period 2006-2100. We apply future land use scenarios following the two RCPs in CMIP5, in which
the extent of crop and pasture cover in the WUS increases by 30% in future climates, with most of
these changes occurring outside the national forest and park lands in the region (Brovkin et al.,
2013; Kumar et al., 2013).

Passive fire suppression results from landscape fragmentation caused by land use (e.g., for

crop and grazing land, roads, and urban areas), and this influence on fire activity is included in the
LPJ-LMfire simulations (Pfeiffer et al., 2013). The model does not, however, consider the active
fire suppression practiced throughout much of the WUS. We therefore limit our study to wildfire
activity on the national forest and park lands of the WUS that are dominated by lightning fires and
where land use for agriculture and urban areas is minimal. To focus only on national forest and
park lands, we apply a 0.5° × 0.5° raster across the WUS that identifies the fraction of each grid
cell that belongs to a national forest or national park (Fig. S3), and we consider only these areas in
our analysis. To calculate fire emissions, we multiply the simulated dry matter burned by the
fraction of national forest or park within each grid cell.
**2.2   Fire emissions**

Fuel biomass in LPJ-LMfire is discretized by plant functional type (PFT) into specific live

biomass and litter categories, and across four size classes for dead fuels. The model simulates
monthly values of total dry matter burned for nine PFTs as in Pfeiffer et al., 2013. To pass LPJ-
LMfire biomass burning emissions to GEOS-Chem, we first reclassify these nine PFTs into the
six land cover types considered by GEOS-Chem. See Table S1 for a summary of the
reclassification scheme. Tropical broadleaf evergreen, tropical broadleaf raingreen, and $C_4$ grasses
are not simulated by LPJ-LMfire in the national forests and parks of the WUS. Emission factors
based on the six land cover types in GEOS-Chem are then applied to dry matter burned from
LPJ-LMfire, resulting in monthly BC and OC emissions over national forests and parks. These
factors are from Akagi et al., 2011. As lightning-started wildfires are dominant over the WUS
forests, an evaluation of fire emissions over national forest and park lands from the LPJ-LMfire
model against the Global Fire Emissions Database (GFED4s) inventory (Giglio et al., 2013) is
included in the Supplement (Fig. S4).
**2.3   GEOS-Chem**

We    use    the    GEOS-Chem    chemical    transport    model    (version    12.0.1;

http://acmg.seas.harvard.edu/geos/). We first carry out a global simulation at 4° latitude x 5°
longitude spatial resolution, and then downscale to 0.5° × 0.625° over the WUS via grid nesting
over the North America domain. For computational efficiency, we use the aerosol-only version of
GEOS-Chem, with monthly mean oxidants archived from a full-chemistry simulation, as described
in Park et al., 2004. Simulations with the fine-scale GEOS-Chem are computationally expensive,
and we first test whether performing five-year simulations will adequately capture the interannual
variability in fire activity generated by the LPJ-LMfire model. We take the average of fire-season
total dry matter burned over five-year time slices in different periods across the 21st century, and
find that these averages differ from the same quantity averaged over ten-year time slices by less
than 20%, which is much less than the discrepancies caused by using different climate models in
future predictions (Sheffield et al., 2013). This relatively small difference gives us confidence that
five-year simulations in GEOS-Chem will suffice for this study. We therefore perform two five-
year time slice simulations for each RCP, covering the present day (2011-2015) and the late-21$^{st}$
century (2096-2100). The GEOS-Chem simulations are driven with present-day MERRA-2
reanalysis meteorology from NASA/GMAO (Gelaro et al., 2017) to isolate the effect of changing
wildfires on U.S. air quality. The simulations include emissions of all primary PM and the gas-
phase precursors to secondary particles, with non-fire particle sources comprising fossil fuel
combustion from transportation, industry, and power plants from the 2011 EPA NEI inventory. In
the future time slices, non-fire emissions remain fixed at present-day levels.

Our study focuses on carbonaceous PM (smoke PM; BC+OC), which are the main

components in wildfire smoke (Chow et al., 2011). For the present day, we apply 5-year (2011-
2015) averaged GFED4s emissions to those regions that fall outside national forests and parks and
temporally changing LPJ-LMfire emissions from the two RCPs within the Forests. Implementing
the combined emissions allow us to further validate the simulated results in this study using
observations from the Interagency Monitoring of Protected Visual Environments (IMPROVE)
network (Figs. S5-S6). For the future time slices, we assume that fires outside national forests and
parks remain at present-day levels, and we again combine the 2011-2015 GFED4s fire emissions
with the temporally changing, future LPJ-LMfire emissions over the national forests and parks.

**3   Results**
**3.1   Spatial shifts in fire activity**

Under both RCPs, 21$^{st}$ century climate change and increasing atmospheric CO$_2$

concentrations lead to shifts in the distribution of total living biomass and dry matter burned. Fig.
1 shows the changes in monthly mean temperature and precipitation averaged zonally over grid
cells at each 1° latitude of the West, relative to the present day, defined as ~2010. Peak temperature
enhancements in JAS occur between 36°-42° N for ~2050 and ~2100 in both RCPs, with a
maximum enhancement of 4 °C for RCP4.5 and 6 °C for RCP8.5 in 2100. Significant decreases
in JAS precipitation occur between 33°-45° N under RCP4.5 and at latitudes north of 39° N under
RCP8.5 for ~2100. The maximum decrease in monthly precipitation over the West is ~40 kg m$^{-2}$
(~60%) in JAS under both RCPs. These warmer and drier conditions favor fire activity under future
climate.

Fires and smoke production are dependent on fuel load, and throughout the 21$^{st}$ century,

total living biomass in the WUS is primarily concentrated in northern forests (Fig. 2). For RCP4.5,
living biomass exhibits significant enhancements in U.S. national forests and parks at latitudes
north of 43° N in the 2050 time slice and north of 45° N in the 2100 time slice. North of 46° N,
the change in living biomass at 2100 (~0.4 kg C m$^{-2}$) is double that at 2050 (~0.2 kg C m$^{-2}$). At
latitudes south of 40°N, living biomass in RCP4.5 is generally invariant over the 21$^{st}$ century. In
RCP8.5, living biomass also increases significantly near the Canadian border – e.g., as much as
~0.2 kg C m$^{-2}$ for the 2050 time slice and ~0.4 kg C m$^{-2}$ for the 2100 time slice, relative to the
present day. In contrast, at latitudes between 42°-47° N in RCP8.5, total living biomass decreases
by as much as -0.6 kg C m$^{-2}$ for ~2100. For both RCPs, these mid-century and late-century changes
in total living biomass are significant ($p < 0.05$) across nearly all latitudes. In RCP4.5, the spatial
shifts of total living biomass are relatively weak from 2050 to 2100, consistent with the moderate
climate scenario with gradual reduction in greenhouse gas emissions after 2050. However, under
the continued-emissions climate scenario RCP8.5, total living biomass in these forests first
increases by 2050 and then decreases by ~10% by 2100, indicating a strongly disturbed vegetation
system due to climate change. Despite this decrease, living biomass in this scenario is still
abundant in the West in 2100, especially over the northern forests (not shown), suggesting that
future climate change will not limit fuel load for fire ignition or spread. Table 1 summarizes these
results.
LPJ-LMfire simulates boreal needleleaf evergreen and boreal and temperate summergreen
(broadleaf) trees as the dominant plant functional types (PFTs) in the national forests and parks of
the WUS; these PFTs together account for ~90% of the total biomass in our study domain. Changes
over the 21st century (Fig. 2) reflect the changes in the growth and distribution of these PFTs, with
increases in living biomass in the north and decreases in the south in both RCP scenarios (Fig. S7).
In the 2100 time slice, vegetation shifts further north than in the 2050 time slice. The reasons for
this shift can be traced to the climate regimes favored by different vegetation types, with temperate
and boreal trees showing moderate to strong inclination in their growth along the north-south
temperature gradient (Aitken et al., 2008). For example, the temperate broadleaf summergreen
PFT favors regions with moderate mean annual temperatures and distinct warm and cool seasons
(Jarvis and Leverenz, 1983), while boreal needleleaf evergreen generally occurs in colder climate
regimes (Aerts, 1995). With rising temperatures, the living biomass of temperate summergreen
trees increases in most states in the WUS, with maximum enhancement of $+1.0\,kg\,C\,m^{-2}$ in western
Washington, northern Montana, and Idaho by 2100 in RCP8.5 relative to 2010. Decreases in this
vegetation type for this scenario occur in the south, as much as $-0.5\,kg\,C\,m^{-2}$ in New Mexico. In
contrast, boreal trees increase in only a few regions in the far north, with a substantial contraction
in their abundance over much of the West, as much as $-4.0\,kg\,C\,m^{-2}$ for boreal needleleaf evergreen
by 2100 in RCP8.5 over the northern forests.
Simulated area burned from lightning-ignited fires in the national forests and parks of the
WUS increases by ~30% by ~2050, and by ~50% by ~2100 for both RCPs (not shown),
comparable to the predicted 78% increase in lightning-caused area burned in the U.S. under a
doubled $CO_2$ climate by Price and Rind, 1994b, which did not account for vegetation changes due
to climate change or changing $CO_2$. That study, however, projected an increase in lightning flashes
and did not consider changing land cover. The changes in area burned we calculate at 2050 are
also within the range of previous studies using statistical methods for this region (e.g., 54% in
Spracklen et al., 2009 and 10-50% in Yue et al., 2013). Fig. 2 further shows that dry matter burned,
a function of both area burned and fuel load, increases relative to the present at most latitudes at
both 2050 and 2100 and in both RCPs. Year-to-year variations in dry matter burned are greater
than those in living biomass due to variations in the meteorological conditions driving fire
occurrence. Previous studies have found that interannual variability in wildfire activity is strongly
associated with regional surface temperature (Westerling et al., 2006; Yue et al., 2013). In our
study, we show that total living biomass mostly decreases at latitudes ~45° N by ~2100 under
RCP8.5, but the peak enhancements in dry matter burned also occur at these latitudes. This finding
indicates that the modeled changes in fire activity are driven by changes in meteorological
conditions that favor fire, as well as by shifts towards more pyrophilic landscapes such as open
woodlands and savannas. As with biomass, lighting-caused fires also shift northward over the 21[st]
century, especially in RCP8.5. In this scenario, dry matter burned increases by as much as 35 g m$^-$
$^2$ mon$^{-1}$ across 40°-48°N at ~2100 compared to the present day. By 2100, the fire-season total dry
matter burned over the forests in the West increases by 24.58 Tg/JAS (111%) under RCP4.5 and
by 50.00 Tg/JAS (161%) in RCP8.5 (Table 1).

The spatial distributions of changes in total living biomass and dry matter burned are shown

in Fig. 3. Under RCP4.5, moderate decreases in total living biomass (by as much as -2.5 kg C m$^{-2}$)
and increases in dry matter burned by 2100 (up to ~70 g m$^{-2}$ mon$^{-1}$) are concentrated in central
Idaho, Wyoming, and Colorado. Large declines in total living biomass and enhancements in dry
matter burned occur in the forests of Idaho and Montana by 2100 under RCP8.5, with a hotspot of
-5.0 kg C $m^{-2}$ in biomass and +100 g $m^{-2}$ $mon^{-1}$ in dry matter burned in Yellowstone National Park.
Similar trends in total living biomass and dry matter burned are also predicted for the Sierra
Nevada (SN) region in California (Fig. S8), with the region defined as in Yue et al., 2014. Predicted
changes in dry matter burned over the SN forests by 2050 are 17-44%, comparable to the calculated
future increases of 30-50% by Yue et al., 2014. We find significant increases in dry matter burned
of 81% by 2100 under RCP8.5 in the SN region. Our results suggest that even as future climate
change diminishes vegetation biomass in some regions of the WUS, sufficient fuel still exists to
allow increases in fire activity and dry matter burned.
**3.2   Smoke PM**
Given the large uncertainty in secondary aerosol formation within smoke plumes (Ortega
et al., 2013), we assume that smoke PM mainly consists of primary BC and OC. We calculate
emissions of fire-specific BC and OC by combining the estimates of the dry matter burned with
emission factors from Akagi et al., 2011, which are dependent on land cover type. Application of
these emissions to GEOS-Chem allows us to simulate the transport and distribution of smoke PM
across the WUS.
With increasing lightning fire activity in most of the national forest and park areas of the
WUS over the 21$^{st}$ century, smoke PM shows modest enhancement for RCP4.5, but more
substantial increases for RCP8.5 (Fig. 4). Smoke PM enhancements in RCP4.5 occur primarily
over the forests along the state boundaries of Idaho, Montana, and Wyoming, with large increases
by as much as ~10 μg $m^{-3}$ in Yellowstone National Park. Scattered increases in smoke PM in
RCP4.5 are also predicted over the forests in northern Colorado, northern California, western
Oregon, and central Arizona. In RCP8.5, smoke PM enhancements are widespread over the
northern states of the WUS by 2100, with significant increases in regions east of the Rocky
Mountains. Increased fire activity and large smoke PM enhancements are seen by 2100 in RCP8.5,
including large areas of the Flathead (Montana), Nez Perce-Clearwater (Idaho), and Arapaho and
Roosevelt (Colorado) National Forests. Particularly large increases – as much as ~40 $\mu$g m$^{-3}$ – occur
in Yellowstone National Park (Wyoming). The increases in fire in these forests significantly
influences air quality over the entire area of Idaho, Montana, Wyoming, and Colorado, with effects
extending eastward to Nebraska and the Dakotas. Increased smoke PM is also predicted over the
Sierra Nevada in both RCPs. In RCP4.5, average smoke PM over the entire WUS increases by 53%
compared to present (Table 1). For RCP8.5, smoke PM more than doubles (109% increase) at
~2100.

**4    Discussion**
We apply an offline, coupled modeling approach to investigate the impact of changes in
climate and vegetation on future lightning-caused wildfires and smoke pollution across the
national forests and parks of the WUS in the 21$^{st}$ century. The GISS model predicts a warmer and
drier climate but nearly constant lightning frequency in both scenarios. For RCP4.5, the late-21$^{st}$
century lightning-caused wildfire-specific smoke PM in the national forests and parks of the West
increases ~53% relative to present. Comparable fire activity between 2050 and 2100 reflect the
effectiveness of the emission reduction strategies after 2050 under RCP4.5, as temperature changes
across the West are relatively flat from 2050 to 2100, with a nearly constant area-averaged mean
annual temperature of ~19.2°C. In RCP8.5, mean annual temperatures continue increasing over
the second half of the 21$^{st}$ century across the West, nearly 2.1°C from 2050, and wildfire-specific
PM concentrations double by 2100. Increased fire activity is driven by changes in meteorological
conditions that favor fire, as well as by shifts towards more pyrophilic landscapes such as open
woodlands and savannas.
In Table 2 we compare predictions in this study with previous fire estimates under future
climate. A difference between these studies and ours is that we consider only changes in fire
activity over the national forests and parks while others examine changes over the whole WUS.
However, we find that in the GFED4s inventory, present-day fire emissions outside these federally
managed areas contribute less than 1% of total DM in the WUS. For area burned, the fraction
outside national forests and parks could be higher than 1%. In contrast, national forests and parks
have abundant fuel supplies, making their fractional contribution to total DM much higher than
would be implied by their fractional contribution to area burned. Also, the fact that lightning is the
dominant driver of wildfire activity over the WUS forests (Balch et al., 2017) allows a reasonable
comparison of the estimates in this study with those in previous studies that include both lightning
and human-started fires over the West.
Table 2 shows that fire activity in the U.S. is predicted to increase in all studies cited. However,
the projected changes in fire metrics such as area burned or in emissions or concentrations of
smoke vary greatly across studies, from ~10-300% relative to present-day values. These
discrepancies arise from differences in the methodologies, fire assumptions, future scenarios
applied, domain and time period considered, and model resolution. The ~80% increases in smoke
emissions that we project by 2050 is generally lower than estimates in previous statistical studies
(e.g., 150-170% in Yue et al., 2013 or 100% in Spracklen et al., 2009). In contrast, the ~80%
increase in smoke emissions in this study at ~2050 are substantially higher than the ~40% increases
predicted by Ford et al., 2018 over the West, though the magnitudes of emission changes in the
two studies are similar. As in our study, Ford et al., 2018 relied on a land cover model, but they
also attempted to account for the influence of future changes in meteorology and population on
the suppression and ignition of fires. Ford et al., 2018 predicted scattered emission increases of
40-45% over the West and a large increase of 85-220% over the Southeast due to increasing
population and the role of human ignition. However, human activities have diverse impacts on
wildfires, and those impacts are a function of land management policy, economics, and other social
trends, making it challenging to predict how trends in human ignitions, fuel treatment, and fire
suppression will evolve in the future (Fusco et al., 2016). In our study, we confine our focus to
fires in national forests and parks in the West, where human activities such as landscape
fragmentation through land use are less important. We further find that the patterns of increasing
fire emissions by 2100 in our study – i.e., over the forests in northern Idaho, western Montana, and
over the U.S. Pacific Northwest – are similar to those predicted by other studies, including Rogers
et al., 2011 and Ford et al., 2018. Our study also predicts significantly elevated smoke PM in Utah,
Wyoming, and Colorado in the late-21$^{st}$ century under RCP8.5 and in regions east of the Rocky
Mountains because of the prevailing westerly winds.

The following limitations apply to our study. The vegetation model simulations of biomass

and fire are driven by meteorology from just one climate model, GISS-E2-R. Over the WUS, this
model simulates future temperature changes at the low end of projections by the CMIP5 ensemble,
making our predictions of future fire conservative (Sheffield et al., 2013; Ahlström et al., 2012;
Rupp et al., 2013). Also, the GEOS-Chem simulations are driven with present-day MERRA-2
meteorology. Besides changes in fire emissions, future work could examine how changing
meteorology may further influence smoke lifetime and transport processes, and investigate the
feedback of fire on meteorology by developing an online coupled modeling approach.
Anthropogenic ignitions are not considered in this study, but fire behavior and therefore burned
area simulated by LPJ-LMfire are primarily governed by meteorology and fuel structure. The fire
simulations are performed on a 0.5°×0.5° grid, which cannot capture some the fine-grain structure
of the complex topography and sharp ecotones present in our study area (e.g., Shafer et al., 2015).
Our study also does not consider the effects of future climate change on the transport or lifetime
of smoke PM, nor the feedback of smoke aerosols on regional climate. Previous work, however,
has shown that climate effects on smoke PM are likely to be small relative to the effect of changing
wildfire activity (Spracklen et al., 2009).
Within these limitations, our results highlight the vulnerability of the WUS to lightning-
caused wildfire in a changing climate. Even though a changing climate decreases the living
biomass in some regions, we find that ample vegetation exists to fuel increases in fire activity and
smoke. Especially strong enhancements in smoke PM occur in the Northern Rockies in the late-
21$^{st}$ century under both the moderate and strong future emissions scenarios, suggesting that climate
change will have a large, detrimental impact on air quality, visibility, and human health in a region
valued for its national forests and parks. Our study thus provides a resource for environmental
managers to better prepare for air quality challenges under a future climate change regime.



**Data availability**
Data related to this paper may be requested from the authors.

**Author contributions**
Y.L. conceived and designed the study, performed the GEOS-Chem simulations, analyzed the data,
and wrote the manuscript, with contributions from all coauthors. J.O.K. performed the LPJ-LMfire
simulations.

**Competing interests**
The authors declare that they have no competing interest.

**Acknowledgments**
This research was developed under Assistance Agreements 83587501 and 83587201 awarded by
the U.S. Environmental Protection Agency (EPA). It has not been formally reviewed by the EPA.
The views expressed in this document are solely those of the authors and do not necessarily reflect
those of the EPA. We thank all of the data providers of the datasets used in this study. PM data
was provided by the Interagency Monitoring of Protected Visual Environments (IMPROVE;
available online at http://vista.cira.colostate.edu/improve). IMPROVE is a collaborative
association of state, tribal, and federal agencies, and international partners. U.S. Environmental
Protection Agency is the primary funding source, with contracting and research support from the
National Park Service. JOK is grateful for access to computing resources provided by the School
of Geography and the Environment, University of Oxford. The Air Quality Group at the University
of California, Davis is the central analytical laboratory, with ion analysis provided by the Research
Triangle Institute, and carbon analysis provided by the Desert Research Institute. We acknowledge
the World Climate Research Programme's Working Group on Coupled Modelling, which is
responsible for CMIP, and we thank the group of NASA Goddard Institute for Space Studies for
producing and making available their GISS-E2-R climate model output. For CMIP the U.S.
Department of Energy's Program for Climate Model Diagnosis and Intercomparison provides
coordinating support and led development of software infrastructure in partnership with the Global
Organization for Earth System Science Portals. The GISS-E2-R dataset were downloaded from
https://cmip.llnl.gov/cmip5/. We thank the Land-use Harmonization team for producing the
harmonized set of land-use scenarios and making available the dataset online at
http://tntcat.iiasa.ac.at/RcpDb/. We also thank X. Yue for providing the raster of southern
California.

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

 **Figures and tables**

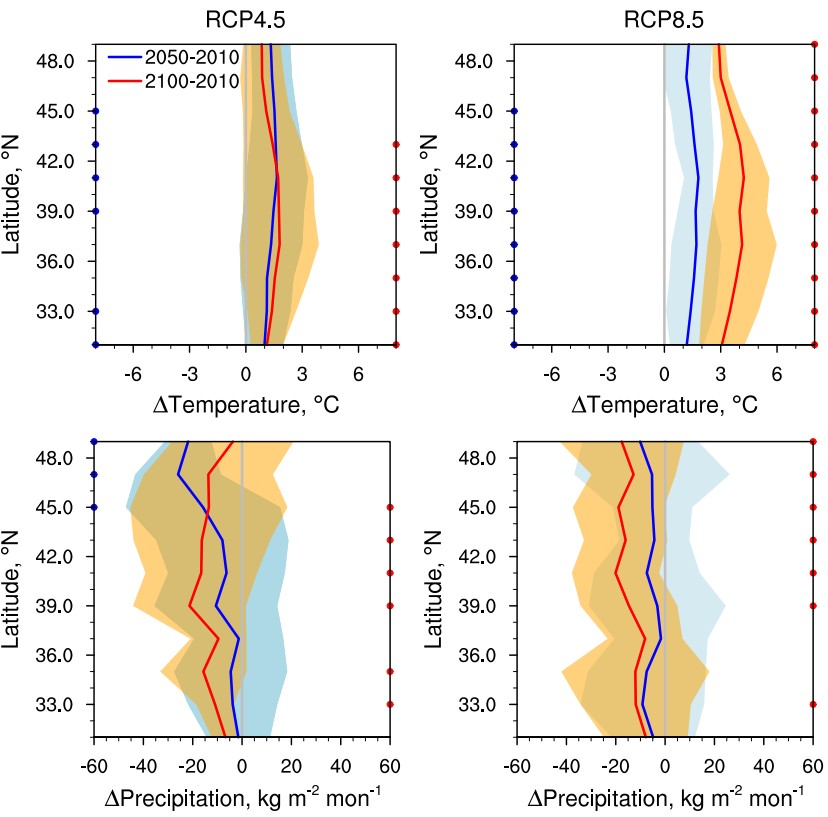


**Figure 1.** Modeled changes in temperature (top) and precipitation (bottom) in July-August-

September (JAS) at ~2050 and ~2100 as a function of latitude over the WUS for RCP4.5 (left)

and RCP8.5 (Nadelhoffer et al.). Changes are zonally averaged and relative to the present day

(~2010), with 5-year averages in each time slice. The bold blue lines show the changes between

2010 and 2050, averaged over all longitudes in the WUS (31°N – 49°N, 100°W – 125°W); bold

red lines show the mean changes between 2010 and 2100. Light blue and orange shadings

represent the temporal standard deviation across the 15 months (5 years x 3 months) of each time

slice. Blue dots along the axes mark those latitudes showing statistically significant differences

between the JAS 2010 and 2050 time slices ($p < 0.05$); red dots mark those latitudes with

statistically significant differences at 2100. Temperatures and precipitations are from the GISS-

E2-R climate model.


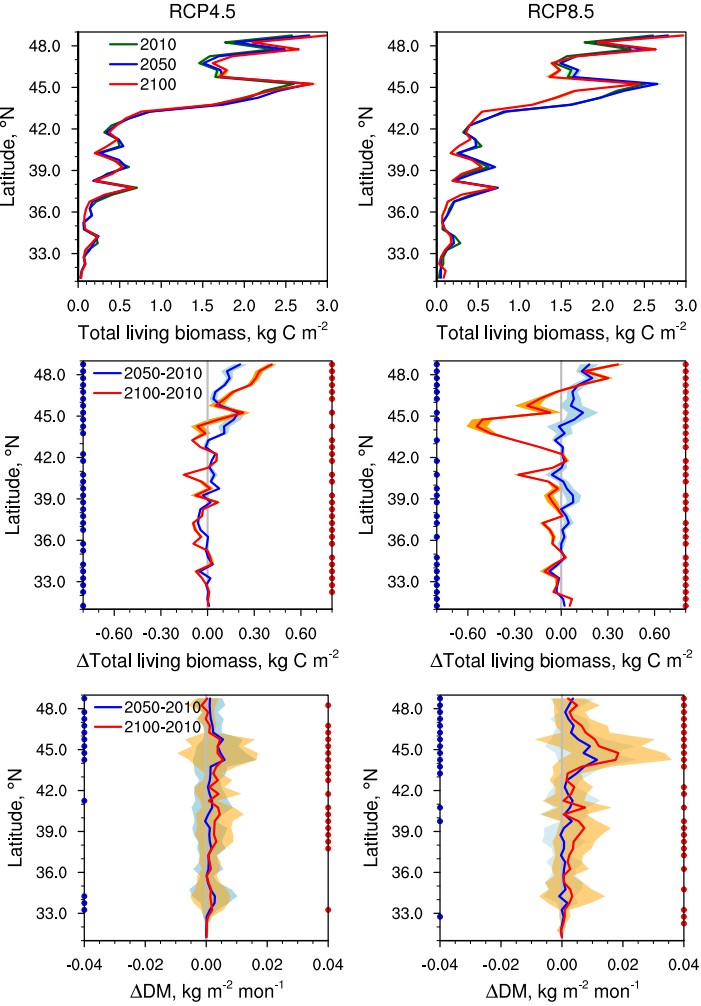


**Figure 2.** The top panel shows total living biomass at ~2010, ~2050 and ~2100 as a function of
latitude over the WUS for RCP4.5 (left) and RCP8.5 (Nadelhoffer et al.), with 5-year averages in
each time slice. The lower four panels are as in Figure 1, but for changes in total living biomass
(middle) and lightning-caused dry matter burned (DM; bottom) as a function of latitude over the
WUS. Results of living biomass and DM are from LPJ-LMfire. As in Figure 1, dots along the axes
mark those latitudes showing statistically significant differences.

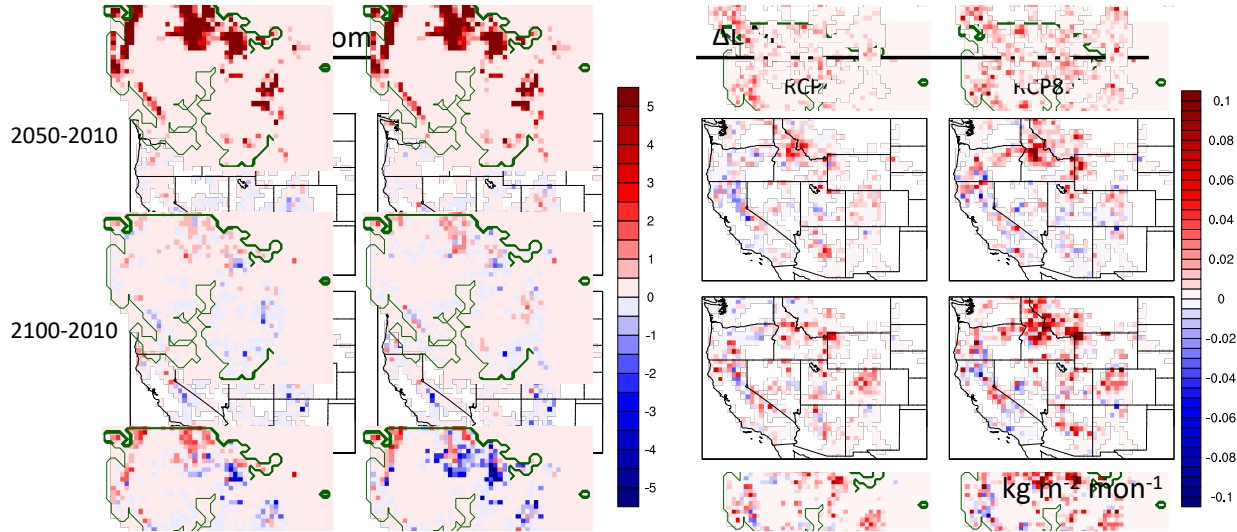

**Figure 3.** Simulated changes in yearly mean total living biomass and monthly mean DM averaged over the fire season in the national forests and parks across the WUS for the RCP4.5 and RCP8.5 scenarios. The top row shows changes between the present day and 2050, and the bottom row shows changes between the present day and 2100. Results are from LPJ-LMfire, with five years representing each time period. The fire season is July, August, and September. White spaces indicate areas outside the national forests and parks.

597

598

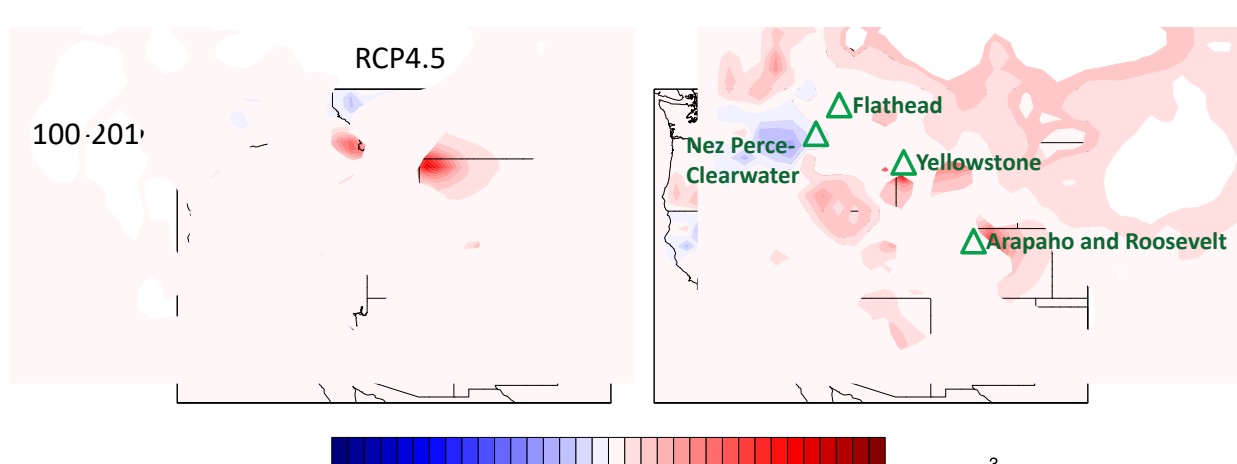

**Figure 4.** Simulated changes in fire-season smoke PM (BC+OC) at ~2100 relative to the present day for RCP4.5 and RCP8.5. Results are from GEOS-Chem at a spatial resolution 0.5° x 0.625°, averaged over July, August, and September. Each time period is represented by a 5-year time slice. National parks and forests that experience large smoke PM enhancements are labeled by green triangles.


**Table 1.** Total living biomass, dry matter burned (DM), and smoke PM (BC+OC) emissions over national forests and parks in the WUS and smoke PM concentrations averaged across the entire West. Values for the present day (~2010) are shown in the top row; changes in ~2050 and ~2100 relative to the present day are shown in bottom two rows. Statistically significant changes are in boldface.

| Time slices | Living biomass[b], Tg/yr | | DM[b], Tg/JAS | | BC+OC emission[b], Tg/JAS | | BC+OC[c], $\mu g\ m^{-3}$ | |
|---|---|---|---|---|---|---|---|---|
| | RCP4.5 | RCP8.5 | RCP4.5 | RCP8.5 | RCP4.5 | RCP8.5 | RCP4.5 | RCP8.5 |
| 2010[a] | 3074.8±33.7 | 3036.9±55.5 | 22.16±4.16 | 30.96±7.15 | 0.15±0.04 | 0.21±0.06 | 2.11±0.48 | 2.55±0.81 |
| 2050-2010[a] | 138.2±46.0 | 126.2±80.2 | **18.0±16.1** | **26.7±14.8** | 0.15±0.13 | **0.23±0.15** | -- | -- |
| 2100-2010[a] | 119.6±34.4 | -270.7±76.1 | **24.6±13.2** | **50.0±18.0** | 0.18±0.14 | **0.39±0.17** | **1.11±1.02** | **2.78±1.73** |

[a] Each time slice represents 5 years; [b] Values are fire-season summations over national forests and parks;
[c] BC+OC concentrations are fire-season averages over the West;
Statistical significance is not calculated for living biomass.

**Table 2.** Comparison of fire predictions in the U.S. under future climate.

| Methods | Region, scenarios, and future time slice | Fire metric and percent increase relative to present day | Smoke PM and percent increase relative to present day | Reference |
|---|---|---|---|---|
| Statistical models for lightning fires | Entire U.S. Doubled $CO_2$ climate | Number of fires: 44% Area burned: 78% | | Price and Rind, 1994b |
| Two climate models | Entire U.S. Doubled $CO_2$ climate ~2060 | Seasonal fire severity rating: 10-50% | | Flannigan et al., 2000 |
| Statistical model | California, U.S. A2 ~2100 | Large fire risk: 12-53% | | Westerling and Bryant, 2008 |
| Statistical models and GEOS-Chem | Western U.S. A1B ~2050 | Area burned: 54% Smoke emission: 100% | Smoke PM concentrations BC: 20% OC: 40% | Spracklen et al., 2009 |
| Climate model with global-scale fire parameterization | Global B1, A1B, A2 ~2100 | Fire occurrence in the western U.S. B1: 120% A1B: 233% A2: 242% | | Pechony and Shindell, 2010 |
| MAPSS-CENTURY 1 dynamic general vegetation model | U.S. Pacific Northwest A2 ~2100 | Area burned: 76-310% Burn severity: 29-41% | | Rogers et al., 2011 |
| Statistical models + GEOS-Chem | Western U.S. A1B ~2050 | Area burned: 63-169% Smoke PM emissions: 150-170% | Smoke PM concentrations: 43-55% | Yue et al., 2013 |
| Statistical models | California, U.S. A1B ~2050 | Area burned: 10-100% | | Yue et al., 2014 |
| Coupled Community Land Model (CLMv4) and Community Earth System Model (CESM) [2] | Western U.S. RCP4.5 and RCP8.5 ~2050 | Smoke PM emissions: • RCP4.5: 100% • RCP8.5: 50% | Total PM$_{2.5}$ concentrations[1] • RCP4.5: 22% • RCP8.5: 63% | Val Martin et al., 2015 |

| | | | | |
|---|---|---|---|---|
| CLMv4.5-BGC with fire parameterization coupled with CESM[3] | Contiguous U.S. RCP4.5 and RCP8.5 ~2050 and ~2100<br><br>Relative to the present day (1995-2005) | Area burned by 2050:<br>• RCP4.5: 67%<br>• RCP8.5: 50%<br>by 2100:<br>• RCP4.5: 58%<br>• RCP8.5: 108% | Total PM$_{2.5}$ concentrations[1] by 2050:<br>• RCP4.5: 146%<br>• RCP8.5: 85%<br>by 2100:<br>• RCP4.5: 108%<br>• RCP8.5: 246% | Pierce et al., 2017 |
| CLMv4.5 with fire parameterization coupled with CESM[3] | Contiguous U.S. RCP4.5 & RCP8.5 ~2050 and ~2100<br><br>Relative to the present day (2000-2010) | Smoke PM emissions by 2050:<br>• RCP4.5: 126%<br>• RCP8.5: 54%<br>by 2100:<br>• RCP4.5: 125%<br>• RCP8.5: 149%<br><br>by 2050 over the West:<br>• RCP4.5: 45%<br>• RCP8.5: 40% | Total PM$_{2.5}$ concentrations[1] by 2050:<br>• RCP4.5: 113%<br>• RCP8.5: 27%<br>by 2100:<br>• RCP4.5: 93%<br>• RCP8.5: 127% | Ford et al., 2018 |
| LPJ-LMfire coupled with GEOS-Chem | Western U.S. RCP4.5 and RCP8.5 ~2050 and ~2100<br><br>Relative to the present day (2011-2015) | Smoke PM emissions by 2050:<br>• RCP4.5: 81%<br>• RCP8.5: 86%<br>by 2100:<br>• RCP4.5: 111%<br>• RCP8.5: 161% | Smoke PM concentrations by 2100:<br>• RCP4.5: 53%<br>• RCP8.5: 109% | This study |

[1] Total PM$_{2.5}$ is the combination of sulfate, ammonium nitrate, secondary organic aerosols, fine
dust, fine sea salt, BC and OC.
[2] This model considers changes in climate, anthropogenic emissions, land cover, and land use.
[3] This model considers changes in climate, anthropogenic emissions, land cover, land use, and
population.