# Peer review of "Trends and spatial shifts in lightning fires and smoke concentrations"

_Atmospheric Chemistry and Physics, 2020_

## Referee Comment (RC1) · Anonymous Referee #1 · 30 Mar 2020

This paper presents future projections of burned area and smoke concentrations from lightning fires on national forest and national park lands in the western US. The paper is generally well written and presents some interesting results. However, I think it could use some clarification before publishing.

Major Suggestions:

- I'd really like a figure that shows specifically the domain that they are looking at with all the national forest and national park lands outlined. This might be the green line on Figure 3, but it is not labeled as such in the caption. Additionally, I think any parks/forests that are mentioned by name in the text (example line 282-283) should have their state
location listed and be labeled on a map (it should not be assumed that all readers know these locations by name).

-I know this will make it wordy and redundant sounding, but I think the authors need to be explicit throughout the paper, every time they mention results, that all their results are only from fires on national park and national forest land in the western US. I think this is especially important in their discussion on smoke concentrations and their comparisons with other studies. It should also be specific in the title.

Line 55- 58 states that one of their aims is to provide results at a higher resolution. I think with this being one of their stated goals, there needs to be more discussion of resolution. They did model simulations at two resolutions, so how do these two resolutions compare? What value does the finer resolution add? How might this finer resolution impact comparisons with other studies?

Minor suggestions:

- It should be "western United States" not "Western United States" throughout the paper. It is incorrect in the title and abstract and switches back and forth throughout the text. I also think national parks and national forests shouldn't be capitalized unless the authors are referring to specific national parks or forests.

- About half-way through the paper, the authors stop using "National Forests and National Parks" and just use "National Forests". I think they should stick with parks as well.

- A flowchart of the modeling set up in the supplement would be beneficial. I found it difficult to follow the input/output of each step in the modeling process. They also need to be clear throughout the text about what each model is actually simulating. For example, they say that LPJ-LMFire simulates meteorology (line 339), but I think they mean that it simulates the effects of meteorology and the meteorology is input. Likewise they say that LPJ-LMFire simulates emissions (line 88), but I think it simulates

area burned, and then they apply the Akagi emission factors to create an emission inventory for GEOS-Chem. (Example: line 39, lightning-caused fire emissions aren't simulated with GEOS-Chem, they are put into GEOS-Chem)

- Table 1 should also have the total BC+OC emissions. I don't think the Dm for the Sierra Nevada needs to be included here. I'd suggest instead adding a supplemental table with several of the large national forests and their results.

- I don't think Table 2 needs to be in the main text.

- I think Table S1 needs to be in the main text since 2 whole paragraphs discuss it.

-Figure S3 is mentioned in the text as an evaluation with GFED4s, nothing about IM-PROVE. I was really confused when I read the acknowledgement section that a large section devoted to IMPROVE when there was no mention of it in the text. This evaluation should be mentioned in the main text, likely under section 2.3.

Line by Line Comments:

Line 19-20: restate that this is for national park and forest lands in the western US.

Line 21-22: This is confusing. Isn't the dry matter burned by lightning-caused fires? A shift in fuel loading could lead to more fires, but if it is already burned, should it not lead to fewer fires?

Line 29-32: Brey et al. (2018) suggests that it is about 30% caused by human ignition in the west. They also note that there are similar drivers for lightning and human caused fires, thus climate changes would likely have a similar impact on both.

Line 35: Studies of what? Be specific.

Line 81: Is a second source missing here (there is a comma and the sentence says "Several studies")? If not, the sentence should read "One study predicted". Also, is there not any more recent papers on lightning and climate change?

[Figure]

Line 83-85: It might be worth noting that this lightning parameterization does not include any potential impacts of aerosols since this work is suggesting an increase in aerosol concentrations.

Line 86-87: I think it would be beneficial to restate this at the end, that lightning isn't increasing, but the area burned from lightning fires is.

Line 91: Is a couple years a long enough spin-up for a vegetation model?

Line 88-94: seems like this should just be in the methods section.

Line 94: Is a five-year time slice long enough to represent the range of interannual variability?

Line 108: What does the "coalescence of fires" mean?

Line115-117: How does the model go from lightning density to fire? Does every lightning strike initiate a fire if there is fuel there?

Line 139: can you use "grid" instead of "raster"? Also, this needs clarification. Is this grid used to create the emissions or just for choosing the analysis area? I'm assuming this is for creating the emissions and the authors use the fraction of the grid box multiplied by the dry area burned and then that gets multiplied by the emission factor to create the emissions to be put into GEOS-Chem? And then for the analysis, do they use any grid box that has any fraction of national park or forest land?

Line 161-162: is this lack of difference for the CTM or LPJ-LMFire and for what variable (20% for emissions seems significant?)?

Line 164-167 should be moved to line 158.

Line 176: why do the GFED4s emissions need to be included at all? If you are just looking at the difference and those are being held the same, it doesn't seem necessary to include them in the simulation at all. Line 178-179 says that they can be compared to observations, but this isn't actually done in the text at all.

Line 196-213: What is causing these increases? Just the warmer climate or is it the shift in biomass type? Does the decrease in precipitation not have a large impact?

Line 213: will not "limit fuel load" for what or with respect to what?

Line 236: changes in what?

Line 243-247: This is a long, confusing sentence.

Line 263: what region? The SN or WUS?

Line 267-268: and the model doesn't simulate this right? Otherwise, you'd need to also include gas-phase precursor emissions calculated for your fire emissions.

Line 273-274: Figure 3 does not show lightning fire activity. It shows changes in dry matter burned and total living biomass.

Line 303-304: be specific that this is for the western US. Also, can you show a map of this, maybe in the supplement? Is this because your area includes any grid box that has any fraction with national park or national forest land? Less than 1% seems really low (protected lands make up <20% of the US)?

Line 312-316: It seems strange to put in the same sentence that there are low smoke emissions compared to some studies, but similar area burned to another study. Do the two studies for the smoke emissions also provide area burned estimates? Otherwise, these should be discussed separately.

Line 308-333: The domain difference and difference in years should be noted along with the difference in resolution.

Line 334-345: Also, there is no feedback of smoke/aerosols on climate included. Also, transport pathways may not vary much, but there are likely some mismatches in the CTM simulation in that the meteorology that is conducive to fires may be more conducive to smoke transport, and the CTM is not using the same input meteorology that was used with LPJ-LMFire.

---

## Referee Comment (RC2) · Alan Wei Lun Lim (Referee) · 7 Apr 2020

Overview

This paper talks about the impacts of future lightning induced wildfires in western United States as projected by a series of computational models. The main model is a fire model that uses future meteorological and land properties as inputs and predicts the occurrences of fires and how much smoke particulate emissions (black carbon and organic carbon) are generated as a result of the fires. Emissions are then used as inputs for a chemistry transport model to predict future impacts on air quality. The paper presents some very interesting results. Parts of the paper lacks specificity, hence

some clarifications are necessary.

Major Suggestions

The authors may want to consider to implement land use changes according to the RCP scenarios in the LPJ-LMfire dynamic vegetation model instead of just assuming 30% increase in cropland and pastures. I understand that anthropogenic effects may be hard to ascertain as per discussed in the paper, but it may be worthwhile to at least look at changes in croplands versus forest cover. For example in RCP4.5: more forests, less crops; RCP8.5: less forests, more crops. Having more cropland in RCP8.5 scenario may lead to more agricultural fires whereas having larger forest cover without human intervention in RCP4.5 scenario may lead to more lightning fires.

I would like to clarify if the model account for agricultural fires? In Table 2, the column for LPJ-LMfire seem to suggest that this fire model does not model agricultural fires although the GEOS-Chem model has a PFT for crops. I guess if the focus of the paper is not about anthropogenic influences on land use changes, and thus lightning fires, then not having this is fine.

It may make the paper more interesting if the authors also list and discuss in greater detail about the possible reasons for the increase in fires, for example, despite having similar lightning activity, stable air and decreased wind led to higher temperatures and hence increasing the occurrences of lightning fires. It may be scientifically interesting to also discuss the most important factor in determining lightning induced fires.

The paper could not discuss any feedback effects of fire on meteorology because the methodology employed simply did not allow such an investigation. Feedback effects of fire on meteorology can be very scientifically interesting, but complicated to investigate. Perhaps this could be future work.

Minor Suggestions

Line 26: I suggest looking at Val Martin, et.al., 2015. Atmos. Chem. Phys., 15, 2805–

2823, 2015. It may be a better cite since it also looks at air pollution and national parks, and is a later research paper.

Line 47: Also check out Li, et. al., 2019. Atmos. Chem. Phys., 19, 12545–12567, 2019 for many different fire models.

Line 81 seems to have a missing citation.

Line 84: A clarification on how the GISS model predicts lightning flashes would be beneficial. Also, only cloud to ground lightning would affect your study. A further clarification on whether cloud to ground lightning remains unchanged throughout the century would be good.

Line 106: It may be necessary to describe in greater detail how each factor in the LPJ-LMfire model affect the predicted fires (incidences of fires, intensity, area burned, etc.) because this is what the whole paper is about.

Line 174: Smoke PM definition should be moved to line 42 to define smoke PM earlier.

Line 291: I would like to suggest a clarification: You are using an offline coupling technique. The present way of phrasing may confuse readers into thinking the fire and atmosphere model are fully coupled.

Supplement Line 24: spelling of lightning

---

## Author Comment (AC1) · 28 May 2020

We thank the reviewers for their insightful comments. Below we provide detailed responses in black, with quotation marks showing the changes made in the manuscript. The line numbers in black refer to the revised (un-tracked) manuscript. The reviewers' comments are in blue.

**Author Response to Reviewer #1**

This paper presents future projections of burned area and smoke concentrations from lightning fires on national forest and national park lands in the western US. The paper is generally well written and presents some interesting results. However, I think it could use some clarification before publishing.

Major Suggestions:

- I'd really like a figure that shows specifically the domain that they are looking at with all the national forest and national park lands outlined. This might be the green line on Figure 3, but it is not labeled as such in the caption. Additionally, I think any parks/forests that are mentioned by name in the text (example line 282-283) should have their state location listed and be labeled on a map (it should not be assumed that all readers know these locations by name).

We added the map of national forest and park fraction in the Supplement (Fig. S3), which specified our domain with all the national forests and parks. We also revised Fig. 3, Fig. S4, Figs. S7-8 to show results in the national forests and parks only.
We added the state locations of the parks and forests as "the Flathead (Montana), Nez Perce-Clearwater (Idaho), and Arapaho and Roosevelt (Colorado) National Forests." Fig. 4 is now updated to denote the locations of these parks and forests.

-I know this will make it wordy and redundant sounding, but I think the authors need to be explicit throughout the paper, every time they mention results, that all their results are only from fires on national park and national forest land in the western US. I think this is especially important in their discussion on smoke concentrations and their comparisons with other studies. It should also be specific in the title.

The title has been changed to "Trends and spatial shifts in lightning fires and smoke concentrations in response to 21$^{st}$ century climate over the national forests and parks of the western United States."
We also now clarify in the discussion that our study focused on fires in the national forests and parks.

Line 55- 58 states that one of their aims is to provide results at a higher resolution. I think with this being one of their stated goals, there needs to be more discussion of resolution. They did model simulations at two resolutions, so how do these two resolutions compare? What value does the finer resolution add? How might this finer resolution impact comparisons with other studies?

We have removed the mention of finer spatial resolution as an aim of the study, and now clarify that the manuscript focuses on the drivers of lightning fires. In Fig. S5 in the Supplement, we provide a comparison of simulated fire-season smoke PM at the resolutions of 0.5° × 0.625° and 4° x 5°. In the supplement we also added:

Supplement, Lines 42-43. "The finer-resolution simulation provides more detailed distributions of fire activity in the WUS, which are of greater utility to environmental managers."

Minor suggestions:

- It should be "western United States" not "Western United States" throughout the paper. It is incorrect in the title and abstract and switches back and forth throughout the text. I also think national parks and national forests shouldn't be capitalized unless the authors are referring to specific national parks or forests.

Done.

- About half-way through the paper, the authors stop using "National Forests and National Parks" and just use "National Forests". I think they should stick with parks as well.

Done.

- A flowchart of the modeling set up in the supplement would be beneficial. I found it difficult to follow the input/output of each step in the modeling process. They also need to be clear throughout the text about what each model is actually simulating. For example, they say that LPJ-LMFire simulates meteorology (line 339), but I think they mean that it simulates the effects of meteorology and the meteorology is input. Likewise they say that LPJ-LMFire simulates emissions (line 88), but I think it simulates area burned, and then they apply the Akagi emission factors to create an emission inventory for GEOS-Chem. (Example: line 39, lightning-caused fire emissions aren't simulated with GEOS-Chem, they are put into GEOS-Chem)

We have added a flowchart of modeling setup (Fig. S1) in the Supplement.
We have also made the following changes to the main text.
Line 367. We now say, "…fire behavior and therefore burned area simulated by LPJ-LMfire are primarily governed by meteorology and fuel structure."
Line 88. we revised the wording as "Combined with emission factors from Akagi et al., 2011, dry matter burned calculated by LPJ-LMfire can be used to estimate natural wildfire emissions of black carbon (BC) and organic carbon (OC) particles, which are then passed to GEOS-Chem, a 3-D chemical transport model, to simulate the transport and distribution of wildfire smoke across the WUS." We also moved this sentence to the method section.
Line 38. "In this study, we project lightning-caused fire emissions and wildfire-specific PM concentrations over the national forests and parks of the WUS in the mid- and late- 21$^{st}$ century, using a dynamic global vegetation model combined with a chemical transport model."

- Table 1 should also have the total BC+OC emissions. I don't think the Dm for the Sierra Nevada needs to be included here. I'd suggest instead adding a supplemental table with several of the large national forests and their results.

We have added BC+OC emissions to the table, following the reviewer's suggestion.  We also removed DM for the Sierra Nevada from Table 1. Large national forests and parks are typically geographically connected, which indicates fire can easily spread from one forest to the nearby forest lands. Therefore, it might make more sense to discuss the changes in fire activity in these forests together.

- I don't think Table 2 needs to be in the main text.

We moved Table 2 to the Supplement as Table S1.

- I think Table S1 needs to be in the main text since 2 whole paragraphs discuss it.

Done.

-Figure S3 is mentioned in the text as an evaluation with GFED4s, nothing about IMPROVE.
I was really confused when I read the acknowledgement section that a large
section devoted to IMPROVE when there was no mention of it in the text. This evaluation
should be mentioned in the main text, likely under section 2.3.

We now mention our use of IMPROVE data in the main text.
Lines 196-198. "Implementing the combined emissions allow us to validate the simulated results
in this study using observations from the Interagency Monitoring of Protected Visual
Environments (IMPROVE) network (Figs. S5-S6)."

Line by Line Comments:

Line 19-20: restate that this is for national park and forest lands in the western US.

Done.

Line 21-22: This is confusing. Isn't the dry matter burned by lightning-caused fires? A
shift in fuel loading could lead to more fires, but if it is already burned, should it not lead
to fewer fires?

Lines 21-22. "RCP8.5 also shows enhanced lightning-caused fire activity, especially over forests
in the northern states."

Line 29-32: Brey et al. (2018) suggests that it is about 30% caused by human ignition in
the west. They also note that there are similar drivers for lightning and human caused
fires, thus climate changes would likely have a similar impact on both.

Brey et al. (2018) suggested that lightning wildfires cause the majority of burned area in the
western U.S., especially during the fire season. Over national forests and parks, Brey et al. (2018)
also showed lightning was the dominant driver of fire ignition. We have added this citation into
our manuscript.
Lines 30-31. "Over the forests of the western United States (WUS), lightning-caused wildfires
account for the majority of burned area (Abatzoglou et al., 2016; Brey et al., 2018)."

Line 35: Studies of what? Be specific.

Lines 36-37. "Not all these studies that attempt to predict future fire activity have accounted for
changing land cover or have distinguished the effects of lightning fire ignitions from human-
started fires."

Line 81: Is a second source missing here (there is a comma and the sentence says
"Several studies")? If not, the sentence should read "One study predicted". Also, is
there not any more recent papers on lightning and climate change?

Fixed.

Line 83-85: It might be worth noting that this lightning parameterization does not include any potential impacts of aerosols since this work is suggesting an increase in aerosol concentrations.

Lines 115-122. "Several studies have predicted future increases in lightning due to climate change (e.g., Price and Rind, 1994a, Romps et al., 2014). However, the relationship between lightning flash rate and meteorology is poorly constrained in models and depends largely on physical parameters such as cold cloud thickness, cloud top height, or convective available potential energy. In our study, lightning strike density for application in LPJ-LMfire is calculated using the GISS convective mass flux following the empirical parameterization of Magi, 2015. Although observations suggest a link between aerosol load and lightning frequency (e.g., Altaratz et al., 2017), we do not consider that relationship here."

Line 86-87: I think it would be beneficial to restate this at the end, that lightning isn't increasing, but the area burned from lightning fires is.

Done.
Line 314. In the discussion, we added "The GISS model predicts a warmer and drier climate but nearly constant lightning frequency in both scenarios."

Line 91: Is a couple years a long enough spin-up for a vegetation model?

We now clarify our method of spin-up.
Lines 140-142: "For each RCP, LPJ-LMfire simulates vegetation dynamics and fire continuously for the period 1701-2100, with monthly resolution. Continuous 400-year simulations allow for sufficient spin-up."

Line 88-94: seems like this should just be in the methods section.

We have moved all the sentences in this paragraph to the method section on line 88 and line 112.

Line 94: Is a five-year time slice long enough to represent the range of interannual variability?

The reviewer raises an important issue.
Lines 177-184. "Simulations with the fine-scale GEOS-Chem are computationally expensive, and we first test whether performing five-year simulations will adequately capture the interannual variability in fire activity generated by the LPJ-LMfire model. We take the average of fire-season total dry matter burned over five-year time slices in different periods across the 21$^{st}$ century, and find that these averages differ from the same quantity averaged over ten-year time slices by less than 20%, which is much less than the discrepancies caused by using different climate models in future predictions (Sheffield et al., 2013). This relatively small difference gives us confidence that five-year simulations in GEOS-Chem will suffice for this study."

Line 108: What does the "coalescence of fires" mean?

By "coalescence," we refer to the merging of fires.
We now more clearly explain how the LPJ-LMfire model simulates fires.

Lines 99-104. "LPJ-LMfire calculates fire starts as a function of lightning ground strikes and ignition efficiency. Not every lightning strike causes fire. The model accounts for the flammability of different plant types, fuel moisture, the spatial autocorrelation of lightning strikes, and previously burned area. As fires grow in size, the likelihood of fire coalescence or merging increases. Fires are extinguished by consuming the available fuel or by experiencing sustained precipitation (Pfeiffer et al., 2013)."

Line115-117: How does the model go from lightning density to fire? Does every lightning strike initiate a fire if there is fuel there?

Lines 99-100. "LPJ-LMfire calculates fire starts as a function of lightning ground strikes and ignition efficiency. Not every lightning strike causes fire."

Line 139: can you use "grid" instead of "raster"? Also, this needs clarification. Is this grid used to create the emissions or just for choosing the analysis area? I'm assuming this is for creating the emissions and the authors use the fraction of the grid box multiplied by the dry area burned and then that gets multiplied by the emission factor to create the emissions to be put into GEOS-Chem? And then for the analysis, do they use any grid box that has any fraction of national park or forest land?

Here we used "raster" to distinguish from "grid cell." The rasters provide information on the fraction in each grid cell that is used to filter and scale the original data.
We now clarify:
Lines 155-156. "To calculate fire emissions, we multiply the simulated dry matter burned by the fraction of national forest or park within each grid cell."
Also, we added the map of national forest and park fraction in the Supplement (Fig. S3).

Line 161-162: is this lack of difference for the CTM or LPJ-LMFire and for what variable (20% for emissions seems significant?)?

Lines 179-184. We clarified "We take the average of fire-season total dry matter burned over five-year time slices in different periods across the 21$^{st}$ century, and find that these averages differ from the same quantity averaged over ten-year time slices by less than 20%, which is much less than the discrepancies caused by using different climate models in future predictions (Sheffield et al., 2013). This relatively small difference gives us confidence that five-year simulations in GEOS-Chem will suffice for this study."

Line 164-167 should be moved to line 158.

Done.

Line 176: why do the GFED4s emissions need to be included at all? If you are just looking at the difference and those are being held the same, it doesn't seem necessary to include them in the simulation at all. Line 178-179 says that they can be compared to observations, but this isn't actually done in the text at all.

We included a comparison with the IMPROVE dataset in the Supplement:
- Lines 38-50. "We compare the GEOS-Chem results against ground-based measurements from the Interagency Monitoring of Protected Visual Environments (IMPROVE) network in the western U.S…."
- Figs. S5-S6.

The reviewer is correct that we do not need GFED4s if we focused on the differences only. But with GFED4s emissions outside national forests and parks, we were able to provide a complete map which could be potentially useful for health studies.

Line 196-213: What is causing these increases? Just the warmer climate or is it the shift in biomass type? Does the decrease in precipitation not have a large impact?

Lines 264-268. "In our study, we show that total living biomass mostly decreases at latitudes ~45° N by ~2100 under RCP8.5, but the peak enhancements in dry matter burned also occur at these latitudes. This finding indicates that the modeled changes in fire activity are driven by changes in meteorological conditions that favor fire, as well as by shifts towards more pyrophilic landscapes such as open woodlands and savannas."
Lines 322-324. In the discussion section, we also added "Increased fire activity is driven by changes in meteorological conditions that favor fire, as well as by shifts towards more pyrophilic landscapes such as open woodlands and savannas."
Our study did not distinguish the impacts of precipitation only. The changes in fire activity are driven by the combined effects of changes in temperature and precipitation.

Line 213: will not "limit fuel load" for what or with respect to what?

We now address this question.
Lines 229-231. "Despite this decrease, living biomass in this scenario is still abundant in the West in 2100, especially over the northern forests (not shown), suggesting that future climate change will not limit fuel load for fire ignition or spread."

Line 236: changes in what?

We clarified as
Line 256. "The changes in area burned we calculate at 2050 are also within the range of previous studies using statistical methods for this region."

Line 243-247: This is a long, confusing sentence.

Fixed.
Lines 264-268. "In our study, we show that total living biomass mostly decreases at latitudes ~45° N by ~2100 under RCP8.5, but the peak enhancements in dry matter burned also occur at these latitudes. This finding indicates that the modeled changes in fire activity are driven by changes in meteorological conditions that favor fire, as well as by shifts towards more pyrophilic landscapes such as open woodlands and savannas."

Line 263: what region? The SN or WUS?

Line 282. "We find significant increases in dry matter burned of 81% by 2100 under RCP8.5 in the SN region."

Line 267-268: and the model doesn't simulate this right? Otherwise, you'd need to also include gas-phase precursor emissions calculated for your fire emissions.

The reviewer is correct.

Line 273-274: Figure 3 does not show lightning fire activity. It shows changes in dry

Fixed.

Line 303-304: be specific that this is for the western US. Also, can you show a map of this, maybe in the supplement? Is this because your area includes any grid box that has any fraction with national park or national forest land? Less than 1% seems really low (protected lands make up <20% of the US)?

Lines 328-329. "However, we find that in the GFED4s inventory, present-day fire emissions outside these federally managed areas contribute less than 1% of total DM in the WUS."

We understand the concern brought up by the reviewer.
Lines 329-332. "For area burned, the fraction outside national forests and parks could be higher than 1%. In contrast, national forests and parks have abundant fuel supplies, making their fractional contribution to total DM much higher than would be implied by their fractional contribution to area burned."

Line 312-316: It seems strange to put in the same sentence that there are low smoke emissions compared to some studies, but similar area burned to another study. Do the two studies for the smoke emissions also provide area burned estimates? Otherwise, these should be discussed separately.

We removed the comparison with area burned here.

Line 308-333: The domain difference and difference in years should be noted along with the difference in resolution.

Lines 338-339. "These discrepancies arise from differences in the methodologies, fire assumptions, future scenarios applied, domain and time period considered, and model resolution."

Line 334-345: Also, there is no feedback of smoke/aerosols on climate included. Also, transport pathways may not vary much, but there are likely some mismatches in the CTM simulation in that the meteorology that is conducive to fires may be more conducive to smoke transport, and the CTM is not using the same input meteorology that was used with LPJ-LMFire.

We have revised the sentence.
Lines 371-372. "Our study also does not consider the effects of future climate change on the transport or lifetime of smoke PM, nor the feedback of smoke aerosols on regional climate."

Lines 363-366. "Also, the GEOS-Chem simulations are driven with present-day MERRA-2 meteorology. Besides changes in fire emissions, future work could examine how changing meteorology may further influence smoke lifetime and transport processes, and investigate the feedback of fire on meteorology by developing an online coupled modeling approach."

**Author Response to Reviewer #2 - Dr. Alan Wei Lun Lim**

This paper talks about the impacts of future lightning induced wildfires in western United States as projected by a series of computational models. The main model is a fire model that uses future meteorological and land properties as inputs and predicts the occurrences of fires and how much smoke particulate emissions (black carbon and organic carbon) are generated as a result of the fires. Emissions are then used as inputs for a chemistry transport model to predict future impacts on air quality. The paper presents some very interesting results. Parts of the paper lacks specificity, hence some clarifications are necessary.

Major Suggestions

The authors may want to consider to implement land use changes according to the RCP scenarios in the LPJ-LMfire dynamic vegetation model instead of just assuming 30% increase in cropland and pastures. I understand that anthropogenic effects may be hard to ascertain as per discussed in the paper, but it may be worthwhile to at least look at changes in croplands versus forest cover. For example in RCP4.5: more forests, less crops; RCP8.5: less forests, more crops. Having more cropland in RCP8.5 scenario may lead to more agricultural fires whereas having larger forest cover without human intervention in RCP4.5 scenario may lead to more lightning fires.

We did indeed implement scenarios of land use change from different RCPs, and we now clarify our methods.
Lines 143-146: "We apply future land use scenarios following the two RCPs in CMIP5, in which the extent of crop and pasture cover in the WUS increases by 30% in future climates, with most of these changes occurring outside the national forest and park lands in the region (Brovkin et al., 2013; Kumar et al., 2013)."
Line 104. "Our study does not consider changes in human-caused fires, including agricultural fires."

I would like to clarify if the model account for agricultural fires? In Table 2, the column for LPJ-LMfire seem to suggest that this fire model does not model agricultural fires although the GEOS-Chem model has a PFT for crops. I guess if the focus of the paper is not about anthropogenic influences on land use changes, and thus lightning fires, then not having this is fine.

The reviewer is referred to the previous response.

It may make the paper more interesting if the authors also list and discuss in greater detail about the possible reasons for the increase in fires, for example, despite having similar lightning activity, stable air and decreased wind led to higher temperatures and hence increasing the occurrences of lightning fires. It may be scientifically interesting to also discuss the most important factor in determining lightning induced fires.

The reviewer suggested very interesting and important topics to look into. Although these topics were beyond the scope of this study, the suggestions provided good guidance for future work. Lines 322-324. In the discussion section, we added "Increased fire activity is driven by changes in meteorological conditions that favor fire, as well as by shifts towards more pyrophilic landscapes such as open woodlands and savannas."

The paper could not discuss any feedback effects of fire on meteorology because the methodology employed simply did not allow such an investigation. Feedback effects of fire on meteorology can be very scientifically interesting, but complicated to investigate. Perhaps this could be future work.

We now mention this direction for future research.
Lines 364-366. "Besides changes in fire emissions, future work could examine how changing meteorology may further influence smoke lifetime and transport processes, and investigate the feedback of fire on meteorology by developing an online coupled modeling approach."

Minor Suggestions

Line 26: I suggest looking at Val Martin, et.al., 2015. Atmos. Chem. Phys., 15, 2805–2823, 2015. It may be a better cite since it also looks at air pollution and national parks, and is a later research paper.

Done.

Line 47: Also check out Li, et. al., 2019. Atmos. Chem. Phys., 19, 12545–12567, 2019 for many different fire models.

Done.
Lines 55-58. "Dynamic vegetation models with interactive fire modeling provide important estimates for long-term and large-scale changes in fire emissions, with most of these models simulating present-day fire emissions within the range of satellite products but failing to reproduce the interannual variability (Li et al., 2019; Hamilton et al., 2018)."

Line 81 seems to have a missing citation.

Done.

Line 84: A clarification on how the GISS model predicts lightning flashes would be beneficial. Also, only cloud to ground lightning would affect your study. A further clarification on whether cloud to ground lightning remains unchanged throughout the century would be good.

The GISS model results archived for CMIP5 does not provide lightning density.
Line 119. "In our study, lightning strike density for application in LPJ-LMfire is calculated using the GISS convective mass flux following the empirical parameterization of Magi, 2015."

It is true that cloud-to-ground lightning is the direct cause of natural wildfires. We now clarify.
Line 124. "LPJ-LMfire scales lightning flashes to cloud-to-ground lightning strikes, which are the portion of total flashes in clouds that directly causes natural wildfires (Pfeiffer et al., 2013). Therefore, cloud-to-ground lightning frequencies are also considered constant during the 21$^{st}$ century."

Line 106: It may be necessary to describe in greater detail how each factor in the LPJLM fire model affect the predicted fires (incidences of fires, intensity, area burned, etc.) because this is what the whole paper is about.

Lines 99-104. "LPJ-LMfire calculates fire starts as a function of lightning ground strikes and ignition efficiency. Not every lightning strike causes fire. The model accounts for the flammability of different plant types, fuel moisture, the spatial autocorrelation of lightning strikes, and previously burned area. As fires grow in size, the likelihood of fire coalescence or merging increases. Fires are extinguished by consuming the available fuel or by experiencing sustained precipitation (Pfeiffer et al., 2013)."

Line 174: Smoke PM definition should be moved to line 42 to define smoke PM earlier.

Fixed.

Line 291: I would like to suggest a clarification: You are using an offline coupling technique. The present way of phrasing may confuse readers into thinking the fire and atmosphere model are fully coupled.

Lines 312. "We apply an offline, coupled modeling approach."

Supplement Line 24: spelling of lightning

Fixed.